# RETHINKING MEMORIZATION IN LLMS: ON LEARNING BY ROTE VS. WITH UNDERSTANDING

## ABSTRACT

Understanding whether and to what extent token sequences generated by large language models (LLMs) are the result of regurgitating memorized training data or are based on meaningful learning of the training data's syntax and semantics has many important implications. In order to cleanly measure and disentangle token recollection by rote (memorization) from generation with understanding, we create an experimental framework that is based on training LLMs over *sequences generated using formal grammars*. Our framework allows us to better understand the interplay between the two types of learning, namely, *by rote* vs. *with understanding*. Using our framework we make several striking observations that hold consistently across different open-source model families (Pythia, Llama, and Mistral): (a) we find that the learning types are at odds with each other during training, *i.e.*, rote learning harms understanding and by developing understanding, models forget previously memorized sequences, (b) we find that *entropy of the training datasets* impacts the ease of learning, with lower entropy datasets being easier to learn with understanding and higher entropy datasets being easier to learn by rote, (c) we highlight the difficulty of determining the type of learning involved in a model based solely on recollecting a training data sequence. Our surprising results have significant downstream implications in the study and usage of LLMs.

## 1 INTRODUCTION

> *"Every teacher knows that there is a profound difference between a student learning a lesson by rote and learning it with understanding, or meaningfully."*
> – Herbert Simon in The Sciences of the Artificial, Third Edition, 1996

The unsupervised training objective of generative models, particularly auto-regressive large language models (LLMs), raises the potential for learning training data both *by rote* (Bender et al., 2021) and *with understanding* Bubeck et al. (2023). Our goal in this paper is to create a framework that enables us to better understand, measure and distinguish the two types of learning that influence the generation (recollection) of next token in LLMs namely, *memorization*, i.e., learning by rote, and *generalization*, i.e., learning with understanding. Such distinction is challenging, but it has many important implications ranging from assessing privacy risks (Biderman et al., 2023a; Carlini et al., 2021) and copyright concerns (Petroni et al., 2019; Reisner, 2023) with training LLMs to developing a foundational understanding of the cognitive abilities of LLMs, i.e., how efficiently they represent, store, and retrieve information.

To illustrate the central challenges and key ideas of this work, let us conduct a thought experiment. Imagine an English speaker and a German speaker commit a paragraph in German to memory. When recollecting the paragraph, do the two speakers rely on rote learning to the same or different extents? Intuitively, the German speaker understands the syntax and semantics of the tokens in the paragraph, while the English speaker sees the paragraph as a sequence of alphabet tokens. Even before reading the paragraph, given some prefix, the former is more likely to predict the next token correctly than the latter. So it stands to reason that the extent of rote learning involved in recollecting the paragraph is higher for the English speaker than the German speaker.

Our thought experiment above highlights the importance of disentangling the two types of learning that are involved in the generation (recollection) of next tokens in the training data. But, how can we isolate and estimate the extents of the two types of learning? In contrast to prior works on

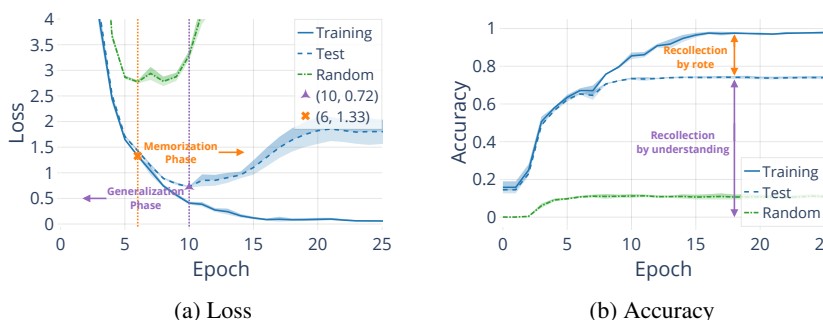

(a) Loss                    (b) Accuracy

Figure 1: **Training and test loss and accuracy for the Llama3-8B model on the hierarchical grammar.** We repeatedly train the model on a set of strings from a hierarchical grammar, and show its performance on the training data, an independently sampled test set and a set of random strings sampled uniformly from the alphabet of the grammar. While the training and test loss decrease similarly in earlier training epochs, the loss of random strings is much higher from the beginning and increases more with training epochs – a demonstration that the LLM indeed learns the grammar by differentiating between grammar strings (train and test data) and non-grammar random strings. The cross-marker shows when the training and test loss diverge (start of learning by rote), and the triangular marker highlights the lowest test loss point (end of learning by understanding).

LLM memorization, our simple but crucial observation is that it is not possible to directly measure a model's memorization. However, it is possible to measure the model's ability to generalize to other *test data*, i.e., sequences with similar syntax and semantics as the training data. The extent to which a model memorized training data can then be indirectly estimated as the difference between its ability to recollect the training data and its ability to generate test data.

To operationalize our ideas, we created a "clean" experimental framework, where we train LLMs to memorize sequences generated by *formal grammars* such as regular and context-free grammars. Formal grammars allow us to study the phenomena of memorization in isolation from generalization. Formal grammars i) allow us to generate training data following the syntax of our choice and guarantee that models have not seen such data during pre-training, ii) enable us to generate test data following the same syntax as training data, which can then be used to estimate the model's generalization and memorization, and iii) give us precise control over all aspects of the data-generation process, such as string length, alphabet size, and entropy. Achieving all of these properties with natural language would not be possible. However, as our findings largely reflect the learning abilities of the LLMs and are not specific to the chosen grammar, we expect them to also apply to natural language data.

The goal of our experiments is to unveil how the two types of learning evolve when LLMs are trained or fine-tuned over datasets. Specifically, we hope to understand the *learning dynamics*, *i.e.* when *memorization begins* and when *generalization ends* throughout repeated exposition to the training data and how the model behaves. Figure 1(a) shows how an LLM's learning evolves when trained over a dataset generated by an example formal grammar. The differences in the progression of loss curves for "training", "test", and "random" sequences show that both types of learning are involved and the interplay between them. Memorization (learning by rote) of training data begins at epoch six; when training and test data curves diverge. Improvement in generalization (learning by understanding) ends at epoch ten; when test data reaches its minimum loss. The extent to which recollection of training data is based on memorization can be estimated by subtracting the recollection that can be attributed to generalization as shown in Figure 1(b).

In order to obtain a comprehensive understanding of the learning processes in LLMs, we perform an extensive set of experiments over pretrained models from different families (Pythia, Llama, and Mistral) with parameter counts spanning more than an order of magnitude. We use data generated from formal grammars different distributions, vocabularies and string structures.

In all cases, we observed similar learning dynamics over repeated training iterations (epochs). We consistently observe that models frequently begin the *memorization phase* before they end the *gen-*

*eralization phase*. But, the exact start and end of the phases as well as their relative impact in generating (recollecting) tokens varies based on the models, training dataset sizes and their underlying grammars. We further analyze the interplay between the two types of learning, motivated by the following questions:

**Q1:** *Can we determine whether and to what extent a trained model has memorized a given training data sequence?* Many prior studies attempted to answer this question based only on the extent to which the trained model can recollect the tokens in the given sequence. We show why it is *impossible* to make such determinations, and how the prior works erred by failing to distinguish cleanly between the two types of learning. Specifically, due to differences in their training, it is possible for two models to recollect tokens in a sequence to the same extent, with one having begun the memorization phase, while the other has not started memorization. Beyond quantifying token recollection, quantifying the memorization of a model requires quantifying its generalization.

**Q2:** *How does learning by rote affect learning with understanding and vice versa?* While we observe that learning by rote and learning with understanding do co-exist, once the generalization phase ends, additional training iterations improve one at the expense of the other. Specifically, after the generalization phase, more memorization comes at the cost of generalization performance. Equally striking, however, is that once a training dataset has been memorized, switching to and training over a new dataset (with the same underlying syntax) leads to *forgetting* of the previously memorized data.

**Q3:** *Are some training datasets easier to generalize or memorize than others?* To address this question, we compare the generalization and memorization performance of models over datasets drawn from grammars with the same syntax and alphabet, but with different entropy values, i.e., different probability distributions over generated sequences. We find that models generalize better on grammars with lower entropy, i.e., they are easier to learn with understanding, but models memorize better on grammars with higher entropy, i.e., they are easier to memorize by rote.

Our work stands in contrast to many related works that solely focused on LLM memorization from the perspective of quantifying privacy risks. Some of the phenomena we unveil as we tackle the above questions are surprising and unexpected. Many have significant implications for quantifying memorization, understanding how memorization works, and estimating the risks of memorizing different types of training data. For many of the observed phenomena, we do not have clear (mechanistic) explanations as of why they happen. But, we feel it is important to report them to the community, as they rule out certain theories related to memorization and can give rise to new ones. We also think that our findings can motivate further studies that will increase our foundational understanding of LLMs' cognitive abilities.

**Related work:** The topic of memorization has received great attention in the context of LLMs that are trained on large "internet-scale" data (Song & Shmatikov, 2019; Carlini et al., 2019; Huang et al., 2022; Zhang et al., 2021; Biderman et al., 2023a; Mattern et al., 2023; Lukas et al., 2023; McCoy et al., 2023). Most of these works propose a definition of memorization to test whether the model can generate a given string (present in the training data) using particular prompts or prefixes. While they subtly differ in how exactly they operationalize a measure of memorization, at a higher level, all these works are concerned with answering the "why" question around memorization, *e.g.* why should memorization be a practical concern? To this end, these works show compelling examples of cases where memorization can hurt (*e.g.* privacy leaks via reconstruction (Carlini et al., 2021) or membership inference (Mattern et al., 2023)). Similarly, there is also a case to be made for memorization being desirable in cases where the goal is to generate facts and reduce LLM hallucinations. Grounding the generation by LLMs in some verified training data sources can be an effective way to generate trustworthy information (Li et al., 2023; Borgeaud et al., 2022; Khandelwal et al., 2019; Tay et al., 2022; AlKhamissi et al., 2022; Petroni et al., 2019; Guu et al., 2020; Haviv et al., 2022).

We differ from existing works in a key aspect. Our goal is to build a foundational understanding of *how* these models memorize and *how* memorization interplays with generalization. Thus, we do not engage with the question of memorization being desirable or undesirable, and rather provide observations on *how* memorization happens at an input-output level. Our work adds to the nascent literature focused on building a better scientific understanding of memorization in LLMs (*e.g.* (Tiru-

mala et al., 2022; Jagielski et al., 2022; Carlini et al., 2022; Kharitonov et al., 2021; Huang et al., 2024; Schwarzschild et al., 2024)).

## 2 PRELIMINARIES AND EXPERIMENTAL SETUP

In order to create a "clean" experimental setup, we use strings generated by probabilistic formal grammars to train and test LLMs. In contrast to natural language, probabilistic grammars provide us with a clearly defined syntax and control over all aspects of the data generation process. A grammar defines a probability distribution over strings. Given an alphabet $T$ for a grammar $G$, the probability distribution $P_G$ over strings from the alphabet $T$ is $P_G : T^* \to [0, 1]$, where $T^*$ is the set of strings generated by $T$. We consider here only grammars that generate finite-length strings. In our experiments, we use training and test sets $D_{\text{train}}, D_{\text{test}} \sim P_G$ sampled from $P_G$ to train and evaluate models.

We call $\ell = |T|$ the size of the alphabet over tokens $T$. The alphabet $T$ of a grammar is a subset of a much larger *vocabulary of all tokens* $V$ used by an LLM, $T \subset V$. Tokens in an LLM's vocabulary can range from single characters to entire words, and its size spans from tens of thousands to a few hundred thousand tokens. In our experiments, we use token alphabets corresponding to lowercase characters in the Latin alphabet and integer numbers. We often refer to the elements of the alphabet as characters, even though they are technically tokens.

In Section 5, we discuss how the entropy of a grammar affects the ability of an LLM to recollect it by rote or with understanding. The entropy $H(G)$ of a grammar $G$ is defined as the entropy of the probability distribution over all strings that can be generated by the grammar (Cover, 1999; Carrasco, 1997); formally

$$H(G) = - \sum_{s \in T^*} P_G(s) \log P_G(s).$$

**Data-generation process.** In our experiments, we use two types of grammars for generating our strings for training and testing: *random* and *hierarchical* grammars, denoted as $G_{\text{random}}$ and $G_{\text{hierarchy}}$, respectively (formal definition is in Appendix A.2). In a random grammar, each token in a string $s$ is independently sampled according to a probability distribution over the alphabet $T$. In the majority of our experiments, we use the uniform probability distribution and sample strings of length $|s| = 64$. We use alphabets $T$ consisting of Latin lowercase characters $T \subset \{a, \ldots, z\}$, with alphabet sizes $\ell \in \{2, 7, 26\}$, and if not stated differently, $\ell = 26$ by default. Note that our random grammar is a probabilistic regular grammar.

The hierarchical grammar is a probabilistic context-free grammar with 21 production rules, grouped into 4 hierarchy levels of non-terminal symbols. Each non-terminal in the grammar has two associated production rules. The grammar has alphabet size $\ell = 9$ over numerical tokens $T = \{1, 2, 3, \ldots, 9\}$. It generates strings $s$ with a fixed length $|s| = 72$, which we sample by expanding the probabilistic production rules. Out of $9^{72} \approx 10^{69}$ possible strings, the grammar generates around $2^{36} \approx 10^{11}$ valid strings. We can change the probability distribution over the generated strings by "skewing" the probabilities of the production rules. In particular, we can increase the probability $p$ of the first production rule for each symbol, while decreasing the probability $1 - p$ of the second production rule. If not stated differently, we use uniform probabilities $p = 0.5$ for the production rules. More details can be found in Appendix A.2 and example strings in Appendix A.3.

**Training and evaluation.** Given a dataset of strings $D$, we train a causal (autoregressive) language model $\mathcal{M}$ on $D$. During training, we minimize the cross-entropy loss for predicting the next token in the string $s \in D$. We denote by $P_{\mathcal{M}}(s_i | s_{[1, i-1]})$ the probability that $\mathcal{M}$ assigns to the token $s_i$ at the $i$-th position of the string $s$ given the prefix tokens $s_{[1, i-1]}$. The cross-entropy loss of the LLM on $D$ is defined as

$$\texttt{Loss}(\mathcal{M}, D) \triangleq -\frac{1}{|D|} \sum_{s \in D} \frac{1}{|s|} \sum_{i=1}^{|s|} \log P_{\mathcal{M}}(s_i \mid s_{[1, i-1]}). \tag{1}$$

In this paper, we study memorization by primarily evaluating models based on their training and test loss $\texttt{Loss}(\mathcal{M}, D_{\text{train}})$ and $\texttt{Loss}(\mathcal{M}, D_{\text{test}})$ on training and test data $D_{\text{train}}$ and $D_{\text{test}}$, respectively. All reported results are aggregates over 5 runs training and test sets sampled with different random seeds We highlight one standard deviation in the plots.

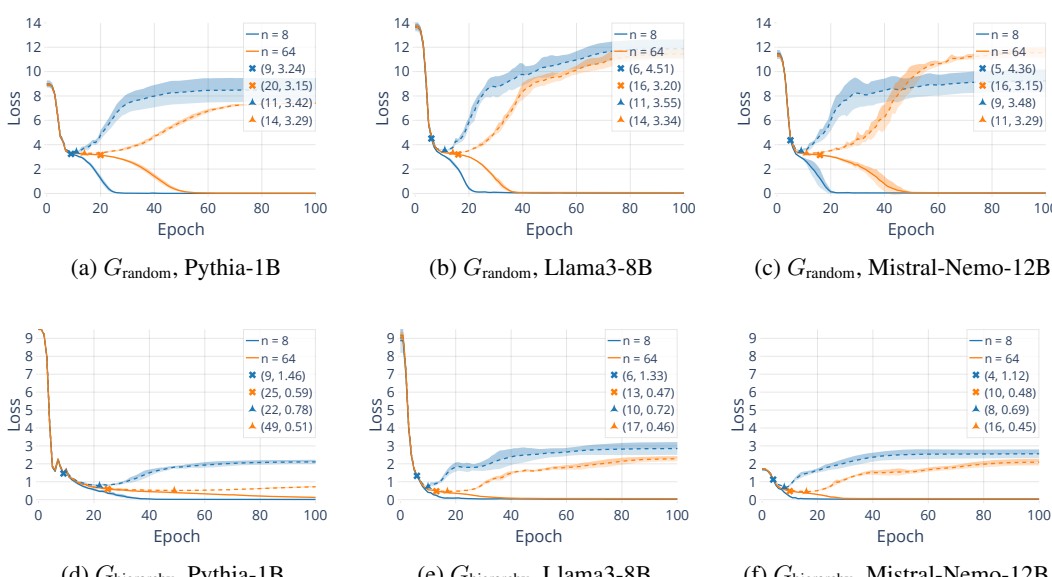

(a) $G_{\text{random}}$, Pythia-1B     (b) $G_{\text{random}}$, Llama3-8B     (c) $G_{\text{random}}$, Mistral-Nemo-12B

(d) $G_{\text{hierarchy}}$, Pythia-1B     (e) $G_{\text{hierarchy}}$, Llama3-8B     (f) $G_{\text{hierarchy}}$, Mistral-Nemo-12B

Figure 2: **Training and test loss for different models on the random and hierarchical grammars.** We show training (solid lines) and test loss (dashed lines) for training dataset sizes of $n = 8$ and $n = 64$. Cross markers indicate the start of memorization (*i.e.* the start of the divergence between training and test loss), and triangular markers the end of generalization (*i.e.* the lowest test loss). Each epoch corresponds to one pass over the dataset, and we fit the entire training set into a single batch (for $n = 64$ via gradient accumulation). We observe two phases: during the *Generalization-Phase* both the training and test loss are similar and decrease together, then they diverge during the *Memorization-Phase* where the training loss decreases further, whereas the test loss increases.

**LLMs:** We use Pythia (Biderman et al., 2023b), Llama-3 (Dubey et al., 2024) and Mistral NeMo[1] families. We use the 1B variant for the the Pythia family, the 8B variant for the Llama-3 family and the 12B variant for Mistral NeMo (only available variant). We refer to each model by its parameter count, *e.g.*, Pythia-1B or Llama3-8B. We choose these models, since they represent popular, modern architectures, and span a wide spectrum of parameter counts (more than an order of magnitude).

## 3 CHARACTERIZING MEMORIZATION AND ITS CONNECTION TO GENERALIZATION

In this section, we first characterize learning dynamics of an LLM in terms of memorization and generalization phases by carefully comparing the LLM's performance on training and test data. Then, we discuss the connection of our work with previous work that attempts to quantify memorization.

### 3.1 DYNAMICS OF MEMORIZATION AND GENERALIZATION

We disentangle learning by rote vs. learning with understanding by characterizing the training dynamics of an LLM at the memorization and the generalization phases, respectively. In particular, we ask two questions, that we empirically answer through training different LLMs on the random and the hierarchical grammars and observing their ability to predict tokens from the training and test datasets generated by the grammar used for their training.

In Figure 2, we see the training and the test loss of different models exposed to different grammars, as a function of the training epochs. We observe that in the initial training epochs, the training and test loss of an LLM decreases similarly. This implies that, albeit never seeing the test data, the LLM demonstrates equal ability to generate the next token regardless of whether the prefix is from the

---

[1]Mistral NeMo blog post

training or the test dataset. Hence, the LLM can transfer its knowledge to the unseen test set. During this period, we say that the LLM is *recollecting tokens with understanding*.

**When does memorization (learning by rote) start?** *We say that the memorization starts at the epoch where the test loss starts diverging from the training loss.* At the point of this divergence, the LLM starts to *recollect tokens by rote*, and it can no longer generate test strings as accurately as training strings. In Figure 2, we highlight the start of the memorization phase with a cross mark. The markers (blue and orange crosses) show the points at which the test loss exceeds the training loss by more than $5\%$, *i.e.* the point after which $\frac{\text{Loss}(\mathcal{M}, D_{\text{test}})}{\text{Loss}(\mathcal{M}, D_{\text{train}})} > 1.05$.

**When does generalization (learning with understanding) end?** As we discussed above, memorization starts at the point where the test loss of the LLM diverges from its training loss. However, the test loss may still decrease in epochs following this point of divergence. *We say that generalization ends at the epoch when the test loss reaches its minimum*; after this epoch, test loss never decreases. In Figure 2, we highlight the points with the lowest test loss with triangle markers.

**Analyzing memorization and generalization phases.** We study the epoch when an LLM reaches (resp. ends) memorization (resp. generalization) and the corresponding test loss for different training sizes across different models and grammars in Figure 2. Our observations are the following:

- **Overlap between the two phases.** In different models and grammars, the generalization phase and memorization phase may overlap (Figure 1a and 2). During these overlapping epochs, the total recollection of an LLM is partly with understanding and partly by rote. It is only *before* the start of memorization, that the LLM learns with understanding, and *after* the end of generalization that the LLM learns by rote.

- **Varying Training Size** Increasing training size results in delayed memorization and longer generalization, and the corresponding test loss is lower – the observation holds across models and grammars. For example, in Pythia-1B model on the random grammar, memorization starts at epoch 9 with training size $n = 8$, compared to epoch 20 with $n = 64$, and the corresponding test loss is 3.24 for $n = 8$ and 3.15 for $n = 64$. For the hierarchical grammar, memorization starts at epoch 9 for $n = 8$, but at epoch 25 for $n = 64$, and the corresponding test loss is 1.46 for $n = 8$ and 0.59 for $n = 64$. A similar observation holds at the end of generalization.

- **Different Models**. The start of memorization and end of generalization may vary among models for the same grammar and training size. For example, in the random grammar and $n = 8$, Pythia-1B, Llama3-8B, and Mistral-Nemo-12B start memorization at epoch 9, 6, and 5, respectively, and end generalization at epoch 20, 16, and 16, respectively. Thus, Pythia-1B model, takes more epochs to start memorization and end generalization compared to other models. The corresponding test loss at these two points of interest also vary among models. These observation leads us to carefully inspect if similar recollection of tokens by different models may occur where one model is in memorization phase while another is in generalization phase, which we discuss next.

### 3.2 QUANTIFYING MEMORIZATION IN PRACTICE

Quantifying memorization and determining whether a model has memorized a string has been the focus of many prior studies (Biderman et al., 2023a; Carlini et al., 2022; Tirumala et al., 2022). All these studies use the recollection accuracy of tokens in order to define their memorization measures. We claim below that such measures do not provide sufficient insight into memorization and may lead to wrong conclusions. Then, we open the floor to discuss some possible alternatives.

**Existing memorization measures.** Carlini et al. (2022) claim that an LLM has memorized a string when it can successfully recollect the next 50 tokens given a prefix. This leads to an *underestimation of memorization*, as it takes a very long time for an LLM to predict correctly 50 consecutive tokens. On the other hand, Tirumala et al. (2022) claim that a *single* token in a string is memorized when an LLM can correctly predict it given some context. This definition is much more liberal, leading to an *overestimation of memorization*, as models can predict many individual tokens correctly via understanding, rather than by rote. Using the right number of tokens to be predicted correctly is a parameter that is hard to set a priori. These examples illustrate that measuring memorization is tricky, as it is easy to be either overly conservative or overly liberal, and it is not clear what the right parameter setting should be.

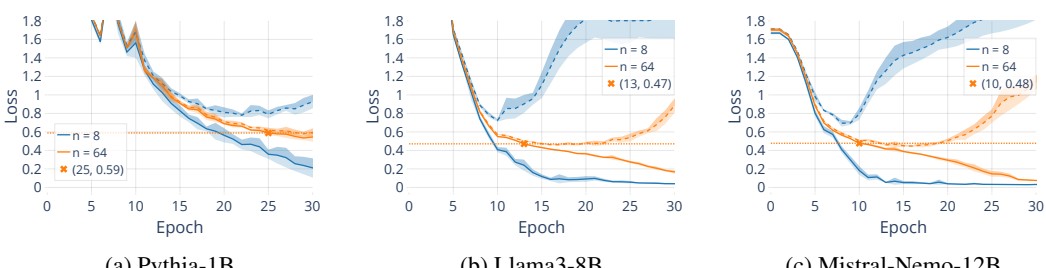

|                | (a) Pythia-1B | (b) Llama3-8B | (c) Mistral-Nemo-12B |
|----------------|---------------|---------------|----------------------|

Figure 3: **Two models can achieve the same loss on a string, with memorization or generalization.** We show a zoomed-in version of the loss on the hierarchical grammar (last row) from Figure 2. Solid lines denote training loss and dashed lines test loss. Horizontal lines show the loss level at which memorization starts for the larger $n = 64$ dataset, and where it intersects with the training loss on the smaller $n = 8$ dataset.

**The impossibility of estimating memorization based on recollection.** In fact, we claim that merely looking into recollection of tokens without differentiating between two modes of learning: learning by rote and with understanding, makes it impossible to decide memorization. To demonstrate this, we show an example of two LLMs that correctly recollect the same amount of tokens, while being in different phases; one is in the generalization and another is in the memorization phase. This example is illustrated in Figure 3. Here, we use two training sizes ($n = 8$ and $n = 64$), and thus, one LLM witnesses more training data than another. We observe that for an identical training loss, such as $0.6$ for Pythia-1B, the LLM trained with $n = 8$ is recollecting via rote (test loss diverges to $\sim 0.8$), while the LLM with $n = 64$ recollects with understanding (test loss remains close to $0.6$) at the point where the $n = 64$ model starts into the *Memorization-Phase*. The same observation holds for all three models. Therefore, *recollection over training strings cannot be the sole indicator to accurately estimate memorization, one needs to differentiate between recollection by rote and with understanding.*

**An alternative quantification of memorization.** Given the above discussion, we claim that in order to quantify memorization one needs to take into consideration both the training loss and test loss of the LLM at any training epoch. Based on this intuition, one candidate measure can be:

$$\texttt{memorization}(\mathcal{M}, D_{\text{train}}, D_{\text{test}}) = 1 - \frac{\texttt{Loss}(\mathcal{M}, D_{\text{train}})}{\texttt{Loss}(\mathcal{M}, D_{\text{test}})},$$

where $\texttt{Loss}(\mathcal{M}, D_{\text{test}}) > 0$ and $\texttt{Loss}(\mathcal{M}, D_{\text{test}}) \geq \texttt{Loss}(\mathcal{M}, D_{\text{train}})$. Intuitively, as long as both losses are similar, the memorization is negligible; memorization increases with an increasing difference between the test and the training loss. Since, in practice, the denominator is always greater or equal to the numerator, our measure takes values in $[0, 1]$.

One can think of the above measure as a suite of measures; Instead of using loss to quantify memorization, other performance metrics (such as accuracy) can also be used with the same high-level interpretation: memorization is the difference of the LLM's performance between training and test datasets computed using appropriate performance metrics.

Such a family of measures come with their own set of shortcomings. These measures depend on the model's performance on train and test data. While we have access to an unseen test set from the same distribution in our experimental setting, it may be more challenging in practical settings as one must generate test and training data to compute such measures. We think that defining and computing such measures in practice will provide new insights regarding the dynamics of memorization of LLMs and leave it as a direction for future work.

## 4 INTERPLAY BETWEEN LEARNING BY ROTE VS. WITH UNDERSTANDING

In the previous section we observed that with repeated exposure to a training set, LLMs first generalize to the distribution of the dataset, before starting to memorize the training data. Importantly,

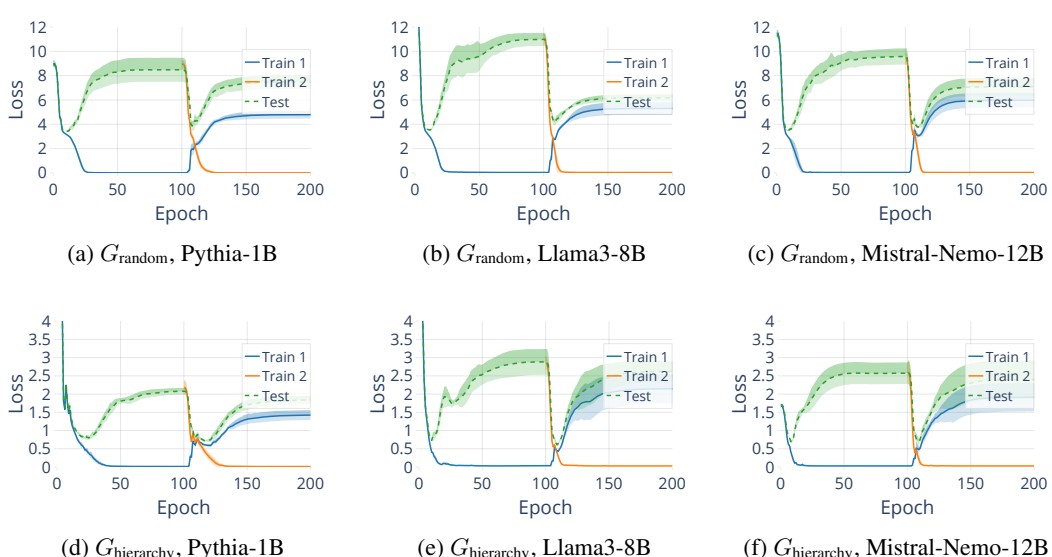

(a) $G_{\text{random}}$, Pythia-1B     (b) $G_{\text{random}}$, Llama3-8B     (c) $G_{\text{random}}$, Mistral-Nemo-12B

(d) $G_{\text{hierarchy}}$, Pythia-1B     (e) $G_{\text{hierarchy}}$, Llama3-8B     (f) $G_{\text{hierarchy}}$, Mistral-Nemo-12B

Figure 4: **Loss for different sets during sequential memorization.** The blue and green curves denote training datasets $D_{\text{train},1}$ and $D_{\text{train},2}$, respectively, that each model is trained on sequentially, and the green curve denotes the test set $D_{\text{test}}$. As models are trained on each dataset, they initially generalize to its distribution, before memorizing it and losing the ability to predict the distribution. Additionally, memorizing a new dataset destroys the ability to recall a previously memorized dataset.

as models enter the *Memorization-Phase*, their loss on the test data increases, *i.e.* their performance on the test data deteriorates. In this section, we investigate the interplay between memorization and generalization in more detail. In particular, we are interested in understanding whether in addition to recall by rote harming recall with understanding, recall with understanding can also harm recall by rote.

To investigate this question, we sample two training sets $D_{\text{train},1}, D_{\text{train},2}$, as well as a test set $D_{\text{test}}$ from distribution $P_G$. Then, we train a model $\mathcal{M}$ first on $D_{\text{train},1}$ and then on $D_{\text{train},2}$, for 100 epochs each, such that it memorizes them one after the other. Additionally, we measure $\mathcal{M}$'s loss on each of the datasets. We use 8 samples for the training datasets, 10 samples for the random test set and 100 samples for the test set from the hierarchical grammar.

Figure 4 shows the results of sequentially training models on different datasets sampled from the random and the hierarchical grammars. For each model $\mathcal{M}$ we observe that as it is trained on the first dataset $D_{\text{train},1}$, its training loss converges to 0, whereas its test loss increases after the end of the *Generalization-Phase*. As we switch and train $\mathcal{M}$ on the second dataset $D_{\text{train},2}$, its loss on $D_{\text{train},2}$ decreases as for $D_{\text{train},1}$. The test loss also initially decreases again, reaching its previous level at the end of the *Generalization-Phase* on $D_{\text{train},1}$, before starting to rise again as $\mathcal{M}$ enters the *Memorization-Phase* on $D_{\text{train},2}$. Interestingly, however, the loss on $D_{\text{train},1}$, which was previously at 0 starts to rise as $\mathcal{M}$ is trained on $D_{\text{train},2}$ and, after plateauing close to the lowest test loss, rises and closely matches the test loss. In other words, as $\mathcal{M}$ memorizes the new dataset $D_{\text{train},2}$, it briefly re-generalizes to $D_{\text{test}}$ and to $D_{\text{train},1}$, before it forgets what it had previously memorized about $D_{\text{train},1}$ while memorizing $D_{\text{train},2}$.

These results show that memorization and generalization are closely linked. Memorizing data from a particular distribution destroys the ability to generalize to that distribution. Conversely, however, (re-)generalization, and memorization of different data also erase previously memorized information. *This result is surprising and points to potential differences between human and machine cognition*: for example, humans can both memorize poems while also being able to write new ones. Our results indicate that this may be more difficult for LLMs.

**Practical Implications.** Our findings here show that we can trigger the forgetting of previously-memorized information, by memorizing new data from the same distribution. For example, one way

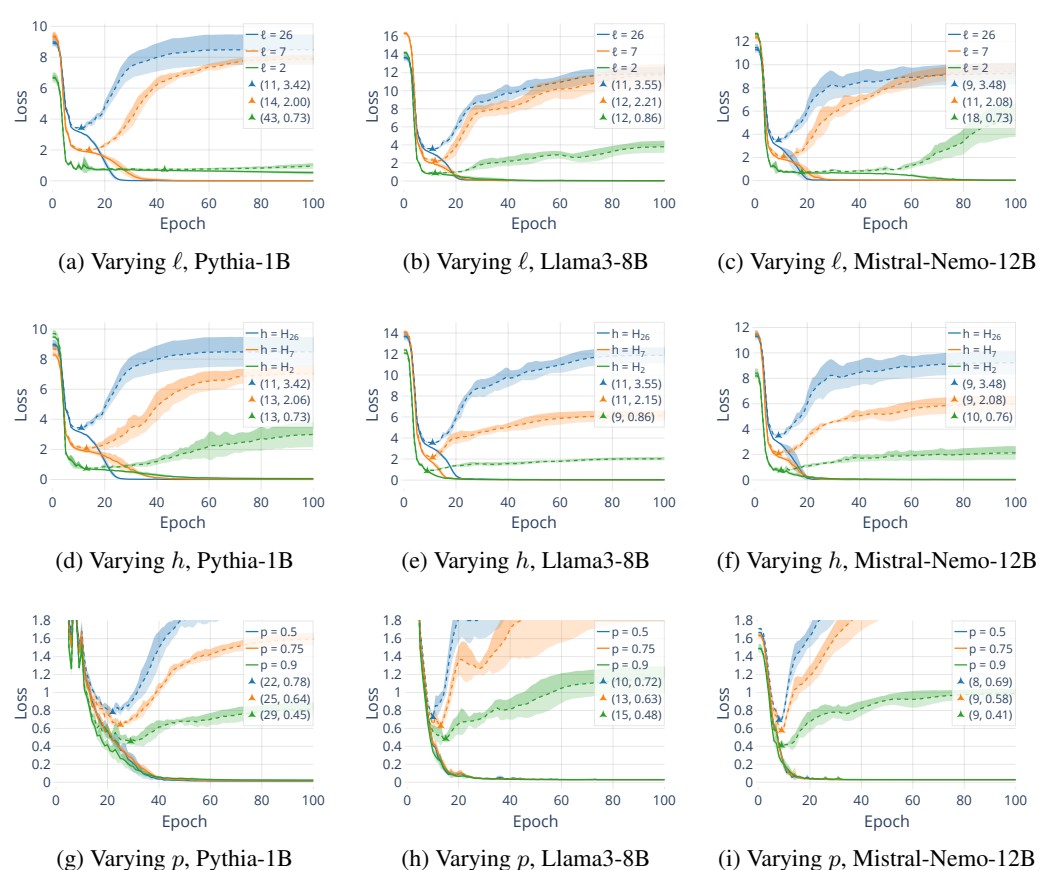

Figure 5: **Loss when varying the entropy of the training dataset.** We vary the entropy of the training data by changing the alphabet size $\ell$ (top row), the entropy level $h$ of the probability distribution by oversampling "a" (middle row), and the skewness $p$ of the hierarchical grammar by modifying the probability of the production rules (bottom row). Solid lines indicate the loss on the training data, and dashed lines the loss on the test data from the same distribution. Models generalize better to lower entropy strings but memorize them more slowly than higher entropy strings.

to make a trained model forget cryptographic keys of a certain format might be to memorize new randomly generated keys with a similar format.

## 5    ARE SOME DATASETS EASIER TO GENERALIZE/MEMORIZE THAN OTHERS?

So far, the distributions $P_G$ induced by the random and hierarchical grammar were the same. An interesting question, however, is how the distribution $P_G$ that data is sampled from affects a model's ability to learn it by understanding and by rote. One factor that is closely associated with memorization is entropy (*i.e.* via compressibility), and here we study how it affects both memorization and generalization. In a nutshell, we make the following observations: 1) the entropy of a grammar's distribution strongly affects a model's ability to learn it by understanding and by rote, and 2) lower entropy distributions are easier to generalize to, but harder to memorize to than higher entropy ones.

To study the impact of entropy on learning, we create distributions based on the random and hierarchical grammars that differ in their entropy. For the random grammar, we change the entropy in two ways: i) we change the size of the alphabet $T$, using sizes $\ell \in 2, 7, 26$, and ii) we oversample the first token ("a"), such that the entropy of the $\ell = 26$ distribution matches that of the $\ell = 2$ ($H_2$) and $\ell = 7$ ($H_7$) distributions. Note that both reducing the alphabet size and oversampling certain tokens from the alphabet reduces the entropy of the distribution based on that alphabet. For the hierarchical

grammar, we skew the probabilities of the production rules to values $p \in \{0.5, 0.75, 0.9\}$. Here, higher values of $p$ correspond to lower entropy. We train models on training sets of size 8 and use test sets of size 10 for the random grammars and 100 for the hierarchical grammars.

**Results.** Figure 5 shows the impact of varying the entropy of the distribution in three different ways, for three different models. As we decrease the entropy of the distribution $P_G$, we consistently observe that models generalize better to the distribution, *i.e.* the lowest test loss they achieve decreases with entropy. On the other hand, strings become easier to memorize as entropy increases. Thus, a lower-entropy grammar is easier to recall by understanding in the generalization phase, but harder to recall by rote in the memorization phase, and vice versa.

**Practical Implications.** Our findings demonstrate that *not all strings are equally memorable or generalizable* – in fact, we show that memorability and generalizability is intrinsically related to the entropy of the distribution from which data is sampled from. Generalizing these findings to natural language strings remains an open challenge, however, as it is unclear how one would estimate the entropy of such strings. Intuitively, however, our findings imply that less typical strings (higher entropy), are easier for an LLM to memorize. Our observations also encourage potential conjectures for how generalization or memorization may be occurring in LLMs – for example, better generalization over lower entropy data could be related to their better compressibility, while better memorization over higher entropy data could be related to length of unique prefixes needed to recollect the next token.

## 6 CONCLUDING DISCUSSION

**Conclusions:** In this paper, we study the phenomenon of learning by rote, *i.e.* memorization, and its connection with learning with understanding, *i.e.* generalization, at a foundational level. We use strings sampled from random and hierarchical formal grammars to create controlled "laboratory" conditions. This data-generation process ensures that 1) the distribution of training data is well-known and identical between training and test sets, and 2) we have full control over the data-generation process, which allows us, for instance, to change the entropy of the distribution.

We make a number of intriguing observations, including that models exhibit a *Generalization-Phase* and a *Memorization-Phase* which overlap, that learning by rote destroys generalization, but also that the inverse is true, and that entropy has a significant impact on the learnability of strings from a distribution; lower-entropy distributions are easier to learn by understanding, but higher-entropy strings are easier to learn by rote afterwards. Based on our results, we argue, that determining the degree of memorization is not possible by measuring only the performance on training strings, since models can achieve the same level of loss on a dataset using both memorization and generalization. Instead, measures of memorization must contrast the recollection performance on the training set with that on an unseen test set from the same distribution.

Our findings have significant implications for studies focusing on quantifying memorization, understanding how memorization works, and estimating privacy risks with memorization. Furthermore, many of our empirical findings cannot be easily explained and the quest for a comprehensive explanatory theory of all our findings raises many open and challenging questions.

**Limitations:** Our insights on memorization rely on synthetic data generated with formal grammars, and it is possible that some of the observations might change for real-world data. We also focus on observing *what* happens during memorization and generalization, and leave it up to future work to study the mechanisms responsible for the observed behavior.

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

# A  ADDITIONAL DETAILS ON THE EXPERIMENTAL SETUP

## A.1  TECHNICAL DETAILS ON THE TRAINING SETUP

**Models:** In this paper, we use pretrained models of the Pythia (Biderman et al., 2023b), Llama-3 (Dubey et al., 2024) and Mistral NeMo[2] families, with 1B, 8B and 12B parameters, respectively. We choose these models, since they represent popular, modern architectures, and span a wide spectrum of parameter counts (more than two orders of magnitude). Pretrained versions for all models are publicly available on the Huggingface Model Hub.

**Training:** We train models to minimize the cross-entropy loss over string $s$ for 100 epochs, with a single batch per epoch, via gradient accumulation. We use a linearly decaying learning rate schedule with a learning rate of $10^{-5}$ for Pythia-1B and $5 * 10^{-6}$ for Llama3-8B and Mistral-NeMo-12B.

## A.2  FORMAL GRAMMAR

A formal grammar, more specifically a probabilistic formal grammar, is defined as a quintuple.

$$G = (N, T, R, S, P)$$

where $N$ is the set of non-terminals, $T$ is the set of terminals (equivalently, tokens), $R$ is the set of production rules, $S$ is the start non-terminal, and $P$ is the set of probabilities on production rules.

**Random Grammar.** We consider a grammar, namely random grammar, to generate random strings. The grammar is an example of a probabilistic regular grammar. The production rules of the grammar are shown below, where $T = \{t_1, t_2, \ldots, t_n\}$, and $N = \{S, A\}$. Each non-terminal has a set of production rules, where the probability of the rule is shown inside the parenthesis. For example, for the non-terminal $A$, $\sum_{i=1}^{n} p_i = 1$.

$$S \to A\ [1]$$
$$A \to t_1\ A\ [p_1]$$
$$A \to t_2\ A\ [p_2]$$
$$\ldots$$
$$A \to t_n\ A\ [p_n]$$

**Hierarchical Grammar.** We consider a hierarchical grammar, which is an example of a probabilistic context free grammar, as shown below. In the grammar, $N = \{S, A7, A8, \ldots, A16\}$ and $T = \{1, 2, 3, \ldots, 9\}$. The grammar has four levels of hierarchy: the non-terminals from top to bottom levels are $\{A16\}$, $\{A13, A14, A15\}$, $\{A10, A11, A12\}$, and $\{A7, A8, A9\}$, followed by terminals $\{1, 2, 3, \ldots, 9\}$. Each non-terminal (except the start non-terminal) has two expansion rules, consisting of non-terminals from the immediate lower level. Further, the expansion rules are probabilistic, where the sum of probabilities of all expansion rules from a given non-terminal is 1. A grammar can be balanced or skewed by varying the probability distribution of expansion rules. In our experiments, we vary the skewness as $p = \{0.5, 0.75, 0.9\}$: the probabilities of the two expansion rules are $p$ and $1 - p$, respectively. Intuitively, $p = 0.5$ generates strings of almost uniform probability, while higher $p$ results in skewing the probability mass to a subset of strings.

---

[2] Mistral NeMo blog post

$$S \rightarrow A16 \ [1]$$
$$A16 \rightarrow A15 \ A14 \ A13 \ [p]$$
$$A16 \rightarrow A13 \ A15 \ A14 \ [1-p]$$
$$A13 \rightarrow A11 \ A12 \ [p]$$
$$A13 \rightarrow A12 \ A11 \ [1-p]$$
$$A14 \rightarrow A11 \ A10 \ A12 \ [p]$$
$$A14 \rightarrow A10 \ A11 \ A12 \ [1-p]$$
$$A15 \rightarrow A12 \ A11 \ A10 \ [p]$$
$$A15 \rightarrow A11 \ A12 \ A10 \ [1-p]$$
$$A10 \rightarrow A7 \ A9 \ A8 \ [p]$$
$$A10 \rightarrow A9 \ A8 \ A7 \ [1-p]$$
$$A11 \rightarrow A8 \ A7 \ A9 \ [p]$$
$$A11 \rightarrow A7 \ A8 \ A9 \ [1-p]$$
$$A12 \rightarrow A8 \ A9 \ A7 \ [p]$$
$$A12 \rightarrow A9 \ A7 \ A8 \ [1-p]$$
$$A7 \rightarrow 3 \ 1 \ 2 \ [p]$$
$$A7 \rightarrow 1 \ 2 \ 3 \ [1-p]$$
$$A8 \rightarrow 6 \ 5 \ 4 \ [p]$$
$$A8 \rightarrow 6 \ 4 \ 5 \ [1-p]$$
$$A9 \rightarrow 9 \ 8 \ 7 \ [p]$$
$$A9 \rightarrow 8 \ 7 \ 9 \ [1-p]$$

## A.3 EXAMPLES OF STRINGS USED IN THE PAPER

| Alphabet and distribution | Strings |
|---|---|
| 2 characters, uniform | bbabbabbababbabaaabbababaaaababb |
| 7 characters, uniform | efceecffdeaggdebbbffddbdabaafaff |
| 26 characters, uniform | pwjqshtulrcxxlpegessmognchaatauv |
| 26 characters, H2 | aaaaaaaaaagaaaaaaaaaaaaaaaaaaaaa |
| 26 characters, H7 | bqadhakmagausabaaaiiaaaaaaaajalp |

Table 1: **[Examples of random strings used in the paper.]** We show the first 32 tokens/characters.

Table 1 shows examples of strings sampled from the random grammar used in the paper. Each character is tokenized individually.

| First production rule probability | Strings |
|---|---|
| $p = 0.5$ | 3126458799873126548793126546451238793 12879... |
| $p = 0.75$ | 6543129876459871231238796546543129873 12879... |
| $p = 0.9$ | 6549873126543129873129876546453129873 12987... |

Table 2: **[Examples of strings sampled from the hierarchical grammar.]**

Table 2 shows examples of strings sampled from the hierarchical grammar used in the paper. Each character is tokenized individually.

