# OpenReview forum: "Rethinking Memorization in LLMs: On Learning by Rote vs. with Understanding"
_ICLR.cc/2025/Conference — Submitted to ICLR 2025_

### Official Review · Reviewer_dqyx · 2024-11-02

**Soundness:** 3
**Presentation:** 2
**Contribution:** 2
**Rating:** 3
**Confidence:** 4

**Summary:**

This paper studies learning curves of various language models on some synthetic languages (generated from a unigram model and a nonrecursive PCFG). They define memorization, or learning by rote, to begin at the point that a gap between the training and test loss begins (more precisely, when training loss > 1.05 * test loss), and generalization, or learning by understanding, to end when test loss is minimized.

-

**Strengths:**

The paper finds that:
- memorization usually begins before generalization ends
- continued training on a different training set restarts generalization and causes forgetting of the first training set
- lower-entropy distributions are easier to generalize on; higher-entropy distributions are more amenable to memorization

**Weaknesses:**

The paper sets for itself the goal of _how_ models memorize (l. 159), but to me this sounds like studying the mechanisms internal to the model that enable it to memorize, whereas this study is a study of learning curves only.

The terms of "learning by rote" and "learning with understanding" are central to the paper. The experiments all center around identifying the beginnings and ends of these two kinds of learning, and one of the aims of the paper is to propose these two definitions as a better alternative to the tests for memorization in previous work. Given the centrality of these two concepts, I would like to see (1) clearer definitions of them earlier in the paper (they are defined only briefly at lines 98-99, and more clearly not until page 6) and (2) some justification for why your definitions of these terms are the right ones.

I'd suggest that you use a single term for each concept, instead of using "learning by rote" and "memorization" as synonyms and "learning with understanding" and "generalization" as synonyms.

The experiments are on synthetic languages instead of naturally occurring data. The reason is to be able to control things like the vocabulary size or the entropy. But there's nothing about the experiments that depends on any details of the data distribution, and the findings are basically the same for the unigram model and the PCFG.

In the end, the findings are not very surprising. I feel that the insights that can be gained by looking at learning curves alone are limited.

**Questions:**

How was the PCFG designed?

---

> ### Author Response · Authors · 2024-11-28
>
> > The paper sets for itself the goal of how models memorize (l. 159), but to me this sounds like studying the mechanisms internal to the model that enable it to memorize, whereas this study is a study of learning curves only.
>
> We indeed focus here on an output level characterization of memorization behvior, without studying intrinsic mechanisms used for memorization by models. We will clarify the framing of the paper in this regard in a revised version.
>
> > The terms of "learning by rote" and "learning with understanding" are central to the paper. The experiments all center around identifying the beginnings and ends of these two kinds of learning, and one of the aims of the paper is to propose these two definitions as a better alternative to the tests for memorization in previous work. Given the centrality of these two concepts, I would like to see (1) clearer definitions of them earlier in the paper (they are defined only briefly at lines 98-99, and more clearly not until page 6) and (2) some justification for why your definitions of these terms are the right ones.
>
> Thank you for the suggestion. We will add clearer definitions in a revised version of the paper, and discuss their motivation.
>
> > I'd suggest that you use a single term for each concept, instead of using "learning by rote" and "memorization" as synonyms and "learning with understanding" and "generalization" as synonyms.
>
> We will precisely define the synonyms early in a revised version of the paper to avoid any confusion.
>
> > The experiments are on synthetic languages instead of naturally occurring data. The reason is to be able to control things like the vocabulary size or the entropy. But there's nothing about the experiments that depends on any details of the data distribution, and the findings are basically the same for the unigram model and the PCFG.
>
> First, the fact that we make consistent observations shows that they are robust.
>
> Second, in order to study training and test loss curves over precisely the same distributions, we need access to a generator of that distribution. We cannot use real-world text datasets for this, since due to their large diversity, we cannot ensure a distribution match between the training and test sets. Additionally, PCFGs allow us to control entropy by modifying the generative probability distributions, and it is unclear how that can be done with natural language data.
>
> > How was the PCFG designed?
>
> We consider two kinds of PCFGs in our experiments, a random grammar for generating random strings and a hierarchical PCFG for generating strings following a hierarchy of non-terminal rules. We discuss details in Appendix A.2.

---

### Official Review · Reviewer_oJUS · 2024-11-03

**Soundness:** 2
**Presentation:** 2
**Contribution:** 1
**Rating:** 3
**Confidence:** 3

**Summary:**

This paper attempts to explore the learning patterns of large language models, specifically whether they learn by rote or by understanding. The authors generate synthesis data via probabilistic context-free grammar(PCFG) as training data and compare loss curves of different LLMs such as Pythia and LLaMA. The main conclusion is that by observing the curves between training loss and test loss, it can infer the learning patterns of the language models. If the training loss continues to decrease while the test loss increases, it is considered to be learning by rote.

**Strengths:**

1.  The topic under investigation is meaningful.

**Weaknesses:**

1. The phenomenon of learning by rote discussed in this paper is similar to overfitting. However, their difference is never explained. If a small model of 128M is pre-trained on WikiText-103 for lots of epochs, a similar phenomenon occurs as well. Essentially, it is overfitting—when the model's capacity exceeds the information provided by the text.  It lacks some quantitative experimental data and rigorous argumentation. For example, the relationship between the amount of information in the pre-train corpus and the model parameters required to alleviate rote memorization, and how to estimate the amount of information. Thus current findings are difficult to constitute actual contributions. The conclusion also lacks practical significance. When the test loss diverges from the training loss, it indicates learning by rote, which is just another name for overfitting.

2. The synthetic data used is too trivial and lacks any distinguishing properties like other works(https://arxiv.org/abs/2305.13673). It simply generates strings based on PCFG, without any extra knowledge or inner logic. To determine whether the model is based on understanding or rote memorization, there needs to be at least some semantic-level properties. For example, executable formal languages. LLMs can understand complex logic, such as sorting and human-defined functions.  For example, making the model fit the results of executable formal languages may help us determine whether the model has achieved true understanding rather than learning by rote.

**Questions:**

Does "training size n=8, n=64" refer to the number of items in the training dataset?

---

> ### Author Response · Authors · 2024-11-28
>
> We thank the reviewer for their valuable feedback. We discuss specific questions in the following.
>
> > The phenomenon of learning by rote discussed in this paper is similar to overfitting. However, their difference is never explained. If a small model of 128M is pre-trained on WikiText-103 for lots of epochs, a similar phenomenon occurs as well. Essentially, it is overfitting—when the model's capacity exceeds the information provided by the text. It lacks some quantitative experimental data and rigorous argumentation. For example, the relationship between the amount of information in the pre-train corpus and the model parameters required to alleviate rote memorization, and how to estimate the amount of information. Thus current findings are difficult to constitute actual contributions. The conclusion also lacks practical significance. When the test loss diverges from the training loss, it indicates learning by rote, which is just another name for overfitting.
>
> We respond to this issue in the global response above.
>
> > The synthetic data used is too trivial and lacks any distinguishing properties like other works(https://arxiv.org/abs/2305.13673). It simply generates strings based on PCFG, without any extra knowledge or inner logic. To determine whether the model is based on understanding or rote memorization, there needs to be at least some semantic-level properties. For example, executable formal languages. LLMs can understand complex logic, such as sorting and human-defined functions. For example, making the model fit the results of executable formal languages may help us determine whether the model has achieved true understanding rather than learning by rote.
>
> We thank the reviewer for their suggestion. While studying semantic properties is an interesting direction, we focus on syntactic properties in this work.
>
> > Does "training size n=8, n=64" refer to the number of items in the training dataset?
>
> Yes. We make similar observations for larger datasets, as discussed in the global response above.

---

### Official Review · Reviewer_qWvw · 2024-11-04

**Soundness:** 3
**Presentation:** 3
**Contribution:** 2
**Rating:** 5
**Confidence:** 3

**Summary:**

The authors study the factor of memorization (rote) and generalization (understanding) during training. The experiments are conducted on training and testing data on synthetic data generated from context-free grammar. The authors propose a new way to quantify memorization, and find that rote learning harms understanding and by developing understanding.

**Strengths:**

* the experiments consider various aspect of the generalization & memorization aspect.

* the new way to quantize memorization feels like a progress. And a revisiting of the traditional bias-variance trade-off seems to be interesting in that it makes the finer-grain distinction that there could be a phase of purely understanding before the traditional trade-off point ("end of learning by understanding").

* Line 418 - 420: this feels like an interesting observation.

**Weaknesses:**

* missing citation and discussion. A more comprehensive study of generalization [2, 3] exists; and CFG data was used for conducting thorough experiments [1] before, and thus is not novel (but the author made them sound novel in abstract Line 16). Also, the second experiment (Figure 4) is closely connected to continual learning, but no discussion of relevant previous results are made.

* some of important point is not addressed in the paper.

Line 73-74: "start of learning by rote" --- Why is learning previous to this not by rote? "end of learning by understanding" --- When is the start of learning by understanding?

* the training set sizes and test sizes are too small. Despite this, the observation is still a bit confusing: train and test loss has small gap, while the author give the impression in Line 196-200 that the output space is large. I think this might be caused by simple CFG grammar. Would be nice to provide some insight for why.

* I disagree with some of the interpretation of results. Or the use of terms by the authors are not rigorous.

Line 422-424: First I think catastrophic forgetting happens in continual learning setting. I don't think the model forgets entirely ("destroy") about D_train,1 as the loss is still lower than D_test.

Line 426 "destroy the ability to generalize...": what's the observation for "destroy"?

Line 428-429 "differences between human and machine cognition": I think this spectulation is taken to far from what's in the paper.

Line 499-500 "Intuitively, .... to memorize": the phrasing seems in-accurate. It's harder to generalize with higher entropy distribution, so the model is **encouraged** to be for the model to be a lookup table. I don't know why it is "easier".

Line 521-522 "measures of memorization must ...": Why is it a "must"?



[1]: Physics of Language Models: Part 1, Learning Hierarchical Language Structures
[2]: COGS: A Compositional Generalization Challenge Based on Semantic Interpretation
[3]: SLOG: A Structural Generalization Benchmark for Semantic Parsing

**Questions:**

* does the syntax support testing hierarchical/compositional generalization?

* line 98: "memorization begins at epoch 6": modern LLM didn't go through data for more than 1 epoch sometimes, how would the author explain the big difference here?

* Line 317: what does it mean by "takes a very long time"? decoding time?

* Line 370-371 "one must generate test and trainign data...": I can't imagine scenario like this. What do the authors mean by "generate"?

* how would the author comment if the defined memorization term (Line 356-358) happens to be negative?

---

> ### Author Response · Authors · 2024-11-28
>
> We thank the reviewer for their insightful comments on the paper, and address them below.
>
> > missing citation and discussion. A more comprehensive study of generalization [2, 3] exists; and CFG data was used for conducting thorough experiments [1] before, and thus is not novel (but the author made them sound novel in abstract Line 16).
>
> We thank the reviewer for pointing us to relevant related work. Compositional generalization [2, 3] focuses on how well a model generalizes to new complex linguistic expressions. As such, [2] puts a constraint on training and test data by considering two different probabilistic grammars for generating them. More specifically, the training set includes systematic gaps that, in the generalization set, must be filled via compositional generalization. [3] extends compositional generalization from the lexical level [2] to structural level by focussing on generalization in the higher level of the CFG hierarchy.
>
> In our paper, we consider the same grammar for evaluating memorization and generalization on training and test data, respectively. Moreover, [2, 3] focus on semantic generalization in a language, while our work investigates syntax – we test how well the LLM memorizes syntax via rote learning vs inferring the underlying grammar for the syntax via generalization. To our best knowledge, no prior study considers CFGs for the disentanglement of the two modes of learning (Section 3.2).
>
>
> The CFGs in our experiments are indeed inspired by [1], who have a different objective of investigating how LLMs learn CFGs by mimicking dynamic programming. We will add the citations in a revised version.
>
> [1]: Physics of Language Models: Part 1, Learning Hierarchical Language Structures
>
> [2]: COGS: A Compositional Generalization Challenge Based on Semantic Interpretation
>
> [3]: SLOG: A Structural Generalization Benchmark for Semantic Parsing
>
> > Also, the second experiment (Figure 4) is closely connected to continual learning, but no discussion of relevant previous results are made.
>
> We will discuss the connection to continual learning in a revised version of the paper.
>
> > Line 73-74: "start of learning by rote" --- Why is learning previous to this not by rote? "end of learning by understanding" --- When is the start of learning by understanding?
>
> Learning by understanding, i.e. the generalization phase, starts right from the first epoch. During the generalization phase, both training loss and test loss are close to each other and the test loss is still decreasing, i.e. the model is still improving its generalization on the test data.
>
> The model starts to learn by rote, i.e. memorize, when its training and test loss start to diverge, since such a divergence can only be due to memorization outperforming its generalization ability.
>
> > the training set sizes and test sizes are too small. Despite this, the observation is still a bit confusing: train and test loss has small gap, while the author give the impression in Line 196-200 that the output space is large. I think this might be caused by simple CFG grammar. Would be nice to provide some insight for why.
>
> The loss gap between training and test data is small only in the initial few epochs during the generalization phase, whereas the gap increases as the LLM enters into memorization. We will include experiments on larger datasets in a future revision.
>
> > I disagree with some of the interpretation of results. Or the use of terms by the authors are not rigorous. Line 422-424: First I think catastrophic forgetting happens in continual learning setting. I don't think the model forgets entirely ("destroy") about D_train,1 as the loss is still lower than D_test.
>
> We will update the phrasing in a revision to indicate that the model retains a small degree of information about the previously memorized strings.
>
> > Line 426 "destroy the ability to generalize...": what's the observation for "destroy"?
>
> We mean the drop in test accuracy. We will clarify this in a revision.
>
> > Line 521-522 "measures of memorization must ...": Why is it a "must"?
>
> We argue that a robust measure of memorization needs to take into account both training and test loss, since it is overwise prone to under- or overestimate the degree of memorization.
>
> > line 98: "memorization begins at epoch 6": modern LLM didn't go through data for more than 1 epoch sometimes, how would the author explain the big difference here?
>
> Even with one epoch of training, strings might be seen multiple times by a model due to duplication in the training data.

---

> ### Author Response · Authors · 2024-11-28
>
> > Line 317: what does it mean by "takes a very long time"? decoding time?
>
> We mean that many training epochs are needed before the model's token prediction accuracy is high enough for sequences of 50 consecutive tokens to be predicted correctly. We will clarify this.
>
> > Line 370-371 "one must generate test and trainign data...": I can't imagine scenario like this. What do the authors mean by "generate"?
>
> By generate we mean sample from the same distribution.
> Sampling is easy when there is access to a data generator, such as with the formal grammars used in this paper.
> However, ensuring that training and test data are sampled from the same distribution is challenging for diverse real-world datasets.
> Hence, accurately estimating memorization on real-world datasets is difficult.
>
> > how would the author comment if the defined memorization term (Line 356-358) happens to be negative?
>
> Training loss, particularly with multiple iterations is pretty much always larger than the test loss.
> Hence, the memorization metric would always be positive in practice.

---

### Official Review · Reviewer_myZZ · 2024-11-05

**Soundness:** 1
**Presentation:** 2
**Contribution:** 2
**Rating:** 3
**Confidence:** 4

**Summary:**

The paper looks at fine-tuning LLMs (Pythia, LLama, Mistral) on strings generated from PCFGs. They analyze the resulting training dynamics and find a generalization phase, and then a memorization phase. These phases are different for different LLMs, and also differ based on whether the training data came from a PCFG or is randomly sampled. Note however that the size of the training data is unusually small and always less than 64 strings.

Next, the paper tries to understand the interplay between memorization and generalization by fine-tuning sequentially on 2 samples $D_1$ and $D_2$,  drawn from the same distribution. They find that training on $D_1$ first causes loss on the test set to go down, and then go up with extended training. After training on $D_1$, training on $D_2$ causes loss to increase on $D_1$, while loss on the test set starts going down again. The authors conclude that they can trigger forgetting of previously memorized examples by memorizing new data from the same distribution, which further can also trigger a round of generalization (since the test set loss also starts going down).

**Strengths:**

- This work looks at the challenging problem of identifying the interplay between generalization and memorization in LLMs, and designs some interesting experiments to study this.
- One interesting experiment is looking at what properties of the data cause memorization, where the authors find increasing entropy of the data distribution increases the model’s propensity to memorize, while lower entropy causes models to enter into the generalization mode.

**Weaknesses:**

- I think the biggest weakness for the paper is that it re-frames many existing and well-known observations from standard machine learning. For instance the memorization / generalization plots are simply the U-shaped val loss plots seen in ML 101 classes. This is commonly referred to as over-fitting. I’m not sure if there’s any deeper insights to be gathered from just that one plot (which the paper heavily relies on).
- Similarly, the fact that loss will increase on a small train set of < 32 examples, after over-fitting to it and then starting to train on a second training sample, is also extremely well-known.
- The size of the datasets used here is unusually small, which makes me wonder if many of the conclusions are an artifact of the small data size.

**Questions:**

To trigger forgetting, a more realistic experiment would be to train a model on a dataset where some examples are specifically marked (with a prefix hash-key). And then check if the model can exactly produce those samples, when prompted with the hash key. Then perform sequential fine-tuning on a new distribution, and see if the model has forgotten to produce previously learnt samples. Crucially, in this setting, make sure that the size of the training set is big (> 10000 samples). Would you still expect to see the same behavior as in Section-4?

---

> ### Author Response · Authors · 2024-11-28
>
> We thank the reviewer for their time and effort in reviewing our paper. We expand the discussion on specific questions in the following.
>
> > I think the biggest weakness for the paper is that it re-frames many existing and well-known observations from standard machine learning. For instance the memorization / generalization plots are simply the U-shaped val loss plots seen in ML 101 classes. This is commonly referred to as over-fitting. I’m not sure if there’s any deeper insights to be gathered from just that one plot (which the paper heavily relies on).
>
> > The size of the datasets used here is unusually small, which makes me wonder if many of the conclusions are an artifact of the small data size.
>
> We respond to these issues in the global response above.
>
> > Similarly, the fact that loss will increase on a small train set of < 32 examples, after over-fitting to it and then starting to train on a second training sample, is also extremely well-known.
>
> While this specific observation may already be known, we make it in Section 4 as part of the larger point that there is a trade-off between generalization and memorization.
>
> > To trigger forgetting, a more realistic experiment would be to train a model on a dataset where some examples are specifically marked (with a prefix hash-key). And then check if the model can exactly produce those samples, when prompted with the hash key. Then perform sequential fine-tuning on a new distribution, and see if the model has forgotten to produce previously learnt samples. Crucially, in this setting, make sure that the size of the training set is big (> 10000 samples). Would you still expect to see the same behavior as in Section-4?
>
> Thanks for the interesting suggestion. We will consider it for a revised version of the paper.

---

### Author Response · Authors · 2024-11-28

We thank all reviewers for their time and feedback. We address the most common issues here and comment on specific points below.

### Our results on memorization match standard notions of overfitting
While we observe similar U-shaped test loss curves as are common in settings where models overfit their training data, we think that memorization in LLMs warrants its own study, and that our results add important nuances.
- First, our work is done in the setting of deep generative sequence models, which substantially complicate the typical overfitting setting where simple models are fitted to classification problems with fixed-sized inputs. More complex models often exhibit new and more complicated behavior, e.g. DNNs exhibit double descend [1], that warrant revisiting previous observations. We show that memorization and generalization are more nuanced than just a specific point at which overfitting starts, because the generalization and memorization phases overlap.
- Second, prior work has shown in the context of generative models that memorization and overfitting are different phenomena [2]. We will include a more thorough discussion of that distinction and that work in a revision of the paper.
- Third, not all our metrics show classical overfitting behavior, e.g. the test accuracy metric in Figure 1 b) only shows a single ascent phase. That shows that the assessment of memorization can depend on the metrics being used, a core claim of our work.

[1] Nakkiran, P., Kaplun, G., Bansal, Y., Yang, T., Barak, B., & Sutskever, I. (2021). Deep double descent: Where bigger models and more data hurt. Journal of Statistical Mechanics: Theory and Experiment, 2021(12), 124003.

[2] van den Burg, G., & Williams, C. (2021). On memorization in probabilistic deep generative models. Advances in Neural Information Processing Systems, 34, 27916-27928.

### The training dataset size is too small
We make very similar observations as for the smaller datasets in the paper on larger ones (> 1000 samples) as well. We will include experiments on larger datasets in a future revision.

### Definitions need to be clearer and better motivated
We will clarify and better motivate the definitions of important terms, particularly memorization and generalization, in a revision of the paper.

---

### Meta-Review · Area_Chair_YAqS · 2024-12-19

**Metareview:**

The paper investigates how large language models (LLMs) learn – do they memorize training data by rote, or do they actually understand the underlying patterns. They use synthetic data generated by formal grammar  to isolate these learning styles. The findings about the trade-off between these learning styles and the impact of data entropy are intriguing. The training datasets used were tiny. This raises serious doubts about the generalizability of the findings. The authors acknowledge this and plan to address it in a revision. Several reviewers pointed out that the observed behavior looked a lot like classic overfitting. The terms "memorization" and "generalization" weren't clearly defined, making the claims less persuasive. The use of PCFGs limits the study to syntactic aspects. The reviewers suggest exploring semantic understanding would be a more significant contribution.

**Additional Comments On Reviewer Discussion:**

The authors' response addresses some weaknesses, particularly by acknowledging the concerns about dataset size and vague definitions and promising revisions. However, the fundamental challenge of directly comparing findings to overfitting remains largely unaddressed.

---

### Decision · Program_Chairs · 2025-01-22

Reject